# Geometric morphometric analysis of spore shapes improves identification of fungi

**Alexander Ordynets** *, **Sarah Keßler, Ewald Langer**

Department of Ecology, Faculty of Mathematics and Natural Sciences, University of Kassel, Kassel, Germany

* a.ordynets@uni-kassel.de

**Data Availability Statement:** Metadata on studied voucher specimens is provided in supporting information S1. DNA sequences newly generated for this study are available from GenBank as accessions MW711723-MW711729. Raw and pre-

## Abstract

Morphology of organisms is an essential source of evidence for taxonomic decisions and understanding of ecology and evolutionary history. The geometric structure (i.e., numeric description of shape) provides richer and mathematically different information about an organism's morphology than linear measurements. A little is known on how these two sources of morphological information (shape vs. size) contribute to the identification of organisms when implied simultaneously. This study hypothesized that combining geometric information on the outline with linear measurements results in better species identification than either evidence alone can provide. As a test system for our research, we used the microscopic spores of fungi from the genus *Subulicystidium* (Agaricomycetes, Basidiomycota). We analyzed 2D spore shape data via elliptic Fourier and principal component analyses. Using flexible discriminant analysis, we achieved the highest species identification success rate for a combination of shape and size descriptors (64.7%). The shape descriptors alone predicted species slightly better than size descriptors (61.5% vs. 59.1%). We conclude that adding geometric information on the outline to linear measurements improves the identification of the organisms. Despite the high relevance of spore traits for the taxonomy of fungi, they were previously rarely analyzed with the tools of geometric morphometrics. Therefore, we supplement our study with an open access protocol for digitizing and summarizing fungal spores' shape and size information. We propagate a broader use of geometric morphometric analysis for microscopic propagules of fungi and other organisms.

## Introduction

In eucaryotic organisms, morphology is an essential source of evidence for biodiversity assessment, taxonomic decisions, and understanding of ecological and evolutionary processes. Morphological information is quickly accessible and can be processed at relatively low costs. Therefore it is broadly used by both researchers and citizen scientists. On the one hand, morphological information provides a starting point for molecular analyses, and on the other hand, serves as reference data to validate the molecular results [1].

Morphology covers two principal concepts: size and shape. The former is easier to acquire, e.g., in the form of linear measurements. However, even if taken in numerous projections, they

processed images and all information extracted from them are available as a published dataset (https://dx.doi.org/10.15156/BIO/807451). The R project with the code, input data, and results of analyses is provided in supporting information S2 and as a repository on GitHub (https://github.com/ordynets/size_vs_shape).

**Funding:** The authors received no specific funding for this work.

**Competing interests:** The authors have declared that no competing interests exist.

provide no information about the geometric structure of the object [2]. The latter concept, the shape, is more challenging in terms of data acquisition and analysis. However, it gains numeric data about the geometric structure and allows reconstruction of the original look of the object [1, 2]. It is a discipline of geometric morphometrics that quantifies a shape and defines it as geometric information about the object that remains after removing the effects of location, rotation, and scale [3].

The toolkit of geometric morphometric analysis depends on the surface properties of the analyzed objects. If distinct points (landmarks) are present that mark angles or cavities and are homologous between the objects, landmark-based methods may be applied. If the landmarks are missing, methods dealing with the complete outline should be used. The most popular in such cases is the Fourier approach that fits the periodic function to the points densely sampled directly from the outline [4]. Among three versions of the Fourier approach, elliptic Fourier analysis is the most advantageous. It decomposes x- and y-coordinates of the outline points and expresses them as functions of the curvilinear abscissa. It does not require equally sampled points, and the outcome can be normalized for size [4, 5].

Fungi are among the species-richest organism groups on Earth [6]. Generally, for morphology-based fungal taxonomy, the features of the disseminated propagules, the spores, are of the highest priority among phenotypic characters [7]. However, there is a difference in how the size and shape of spores are usually treated. The spore size is routinely used for species delimitations, mainly as the linear measurements of length and width [8, 9]. Spore shape, in most cases, is treated as a qualitative trait. Dozens of terms are used based on common geometric shape categories (e.g., "globose" or "cylindric") or similarity to some natural or cultural objects (e.g., "ovoid", "filiform") [10]. The picture becomes even more complicated when subcategories are arbitrarily introduced, e.g., via adding prefix "sub-" or epithets "slightly" or "almost". This qualitative approach prevents an assessment of the trait variation on different levels (individual, populational, species). It also generally hinders the reproducibility of taxonomic research.

In the study of somewhat larger objects than fungal spores (genitalia in Lepidoptera), morphometric analyses resulted in more accurate taxon identification than subjective visual judgments [11]. Furthermore, the most accurate identifications were achieved for a combination of landmark-based geometric information and linear measurements. This is likely due to complementing two mathematically independent sources of evidence [2, 11]. A digitized outline provides even richer geometric information than the landmark configuration. This study hypothesizes that combining linear measurements with geometric information on the outline results in better species identification than either evidence alone.

As a test system for our study, we use the spores of fungal species from the genus *Subulicystidium* Parmasto (Hydnodontaceae, Trechisporales, Agaricomycetes, Basidiomycota, Fungi). We own a large set of spore images for this genus after our previous taxonomic study [12]. There, we failed to distinguish some genetically distinct species based on traditional morphometric analysis. With high diversity of spore shapes, *Subulicystidium* is representative for many fungal lineages but also other microscopic organisms, to which our results and protocol can be applied. An advantage of our test system is the availability of DNA sequences for all studied specimens and a detailed genus-level phylogenetic tree. The DNA-based species assignments serve as reference information to compare shape versus size data performance for automated species identification.

Besides specific approaches for diatom algae, there are only sparse instructions for processing geometric information from microscopic objects [1]. For this reason, we develop a protocol that can be applied to fungal spores as well as other microscopic objects. Our protocol uses software developed for shape analysis of macroscopic objects and additionally involves

general-purpose software for statistical analysis and visualization. Unfortunately, there were few geometric studies of fungal spores and no quantitative comparison between shape and size descriptors [13, 14]. Therefore, we also explored the statistical properties of different shape and size descriptors.

## Methods

### DNA data

We used a balanced set of 30 herbarium specimens which included ten species of *Subulicystidium* each represented by three specimens. We treated two clades of *S. perlongisporum* described in [15] as two separate species, "*S. perlongisporum* 1" and "*S. perlongisporum* 2". For all 30 specimens, we isolated and sequenced the internal transcribed spacer (ITS) of the nuclear ribosomal DNA as described in protocol 1 in [15] or used the sequences we produced earlier [12]. Raw sequence data were processed in Geneious version 5.6.7 [16]. Edited DNA sequences were imported to R version 4.0.3 [17] with the package "Biostrings" version 2.58.0 [18]. Multiple sequence alignment was performed with the package "msa" version 1.18.0 [19] using the MUSCLE algorithm [20] and other settings as default. The alignment ends were then trimmed with the package "ips" version 0.0.11 [21] to the length when 15 of 30 sequences had non-ambiguous base characters in the first and last position of the alignment. The resulting alignment had 734 nucleotide positions. We used the alignment viewer available in the package "ape" version 5.4.1 [22] to check visually the sequence alignment. The best-fitting nucleotide substitution model "TrN+G+I" was chosen according to Akaike information criterion [23]. We searched for the best-scoring maximum likelihood phylogenetic tree with the nearest neighbor interchange strategy and performed bootstrap analysis (1000 replicates) with the "phangorn" package version 2.5.5 [24]. The result was visualized with the R package "ggtree" version 2.4.1 [25].

The newly generated DNA sequences were submitted to GenBank [26]. The complete list of the used DNA sequences with GenBank accessions and metadata on voucher specimens is provided in S1 Table. The studied specimens belong to the herbaria ARAN, CWU, GB, LY, O, and KAS (abbreviations according to [27]).

### Morphological data

**Spore terminology.** To describe the spore morphology, we used the terms as they are found in [28] and [29]. The spores in *Subulicystidium*, as in all Basidiomycota, are produced externally on a meiosporangium called basidium and remain attached to it till they become mature and ready for discharge (Fig 1). It is the proximal part of the spore that directly contacts a horn-like structure, called sterigma, on top of the basidium. The distal part of the spore is found on the opposite side of its long axis. The spore has an adaxial side, i.e., turned to the main axis of basidium, opposite its abaxial side. On the proximal part of the spore, there is a projection called hilar appendix, or apiculus, that is involved in the spore discharge from a basidium [28]. Observing the hilar appendix on the adaxial side of the spore means the spore is seen in the lateral projection. Observing the hilar appendix directly on the main axis of the spore means the spore is seen in the frontal projection. It is correct to compare the shapes of the spores within the same projection [29]. Our study focuses on the spore's lateral face, which is more informative in the case of *Subulicystidium*.

**Image acquisition and pre-processing.** We acquired and pre-processed images from light microscopy as described in detail in our online protocol 30, https://doi.org/10.17504/protocols.io.bdeii3ce]. This paper highlights all essential steps and illustrates the workflow of image processing in Fig 2. We performed all work on images on a desktop computer with

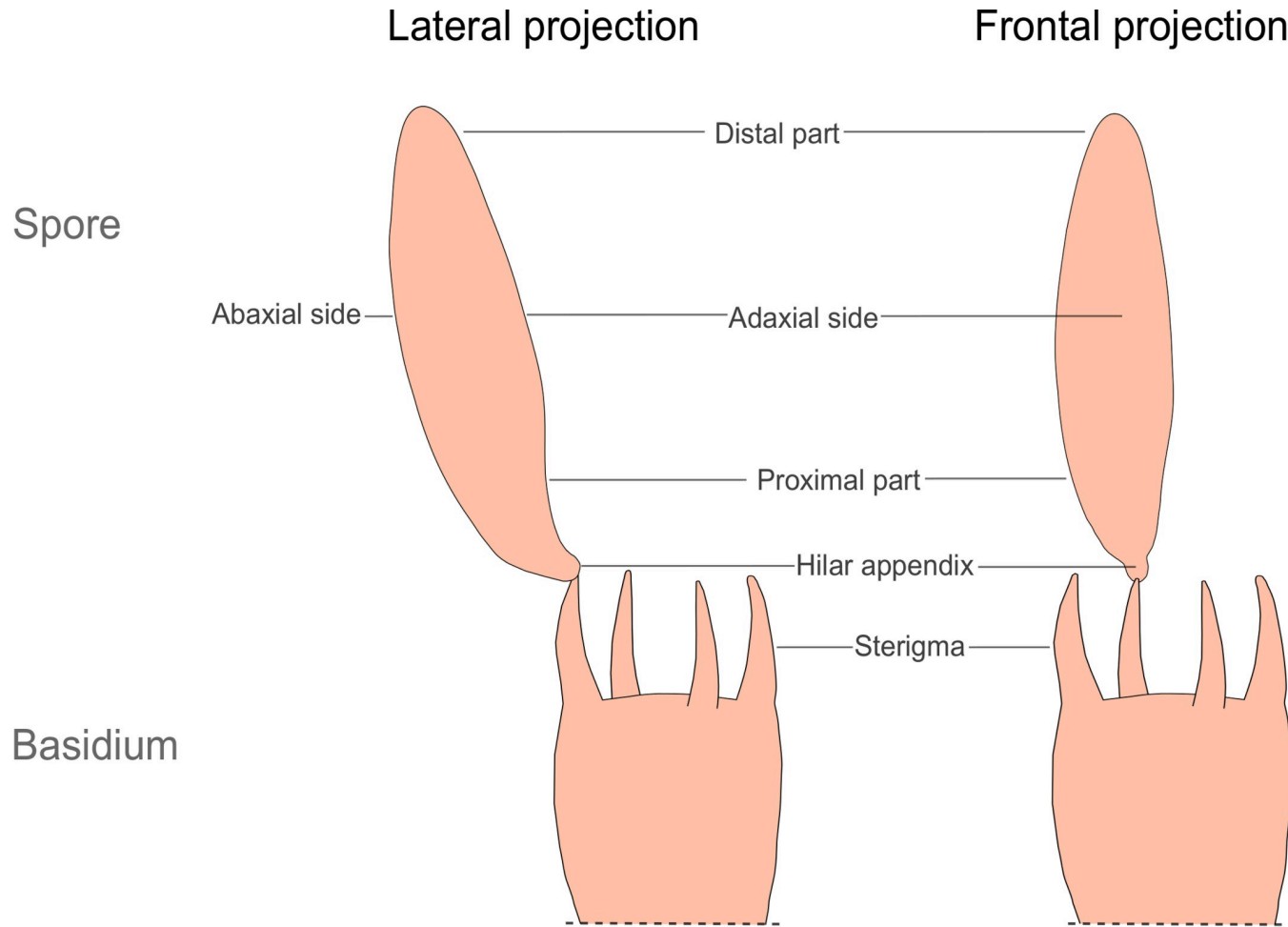

**Fig 1. Crucial terms for describing a spore of the member of Agaricomycetes.** For simplicity, only the upper parts of basidia and only one spore per basidium instead of four are shown.

64-bit Windows 10 operating system (build 19041). The images of spores were obtained from squash preparations of fungal herbarium specimens examined at 1000× magnification (Fig 2A). A 3% aqueous solution of potassium hydroxide (KOH) mixed with a 1% aqueous solution of Phloxine was used as a mounting medium. The captured images (JPEG files) were 1024 × 768 or 2048 ×1536 pixels, while the resolution was always 96 dpi. Bulk image renaming was performed with Bulk Rename Utility version 3.3.1.0 [31] and conversion from JPEG to BMP graphic format with ImageMagick version 7.0.10-Q16 [32].

For further processing, we selected those images that contained one or several healthy (not broken, with intact cell wall) mature (not attached to basidia) and well-focused spores [30]. We also selected only the spores that were lying clearly in the lateral projection. Furthermore, for meaningful shape analysis, spores had to be alignable, i.e., with hilar appendix stretching out in the same direction if the spores were placed in the same orientation. Some images had to be mirrored to meet this criterion. In some cases, we applied additional manual adjustments, namely painted white lines around some parts of the spore outline, to enhance the clarity of the latter and to ensure the absence of contact with the structures that are other spores or not spores (crystals or hyphae), similar to [9] and [33]. These adjustments did not affect the

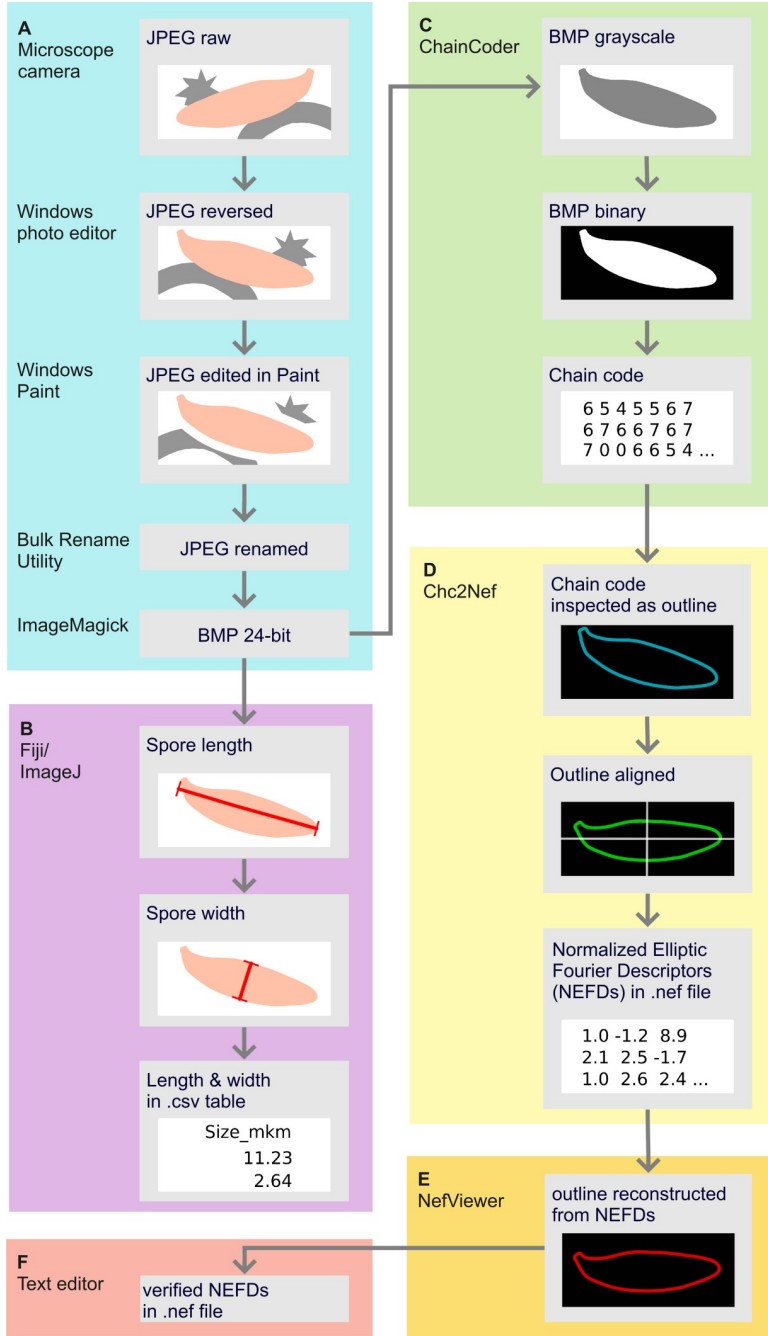

**Fig 2. Workflow showing the extraction of the shape and size information from a fungal spore.** (A) Image capture and pre-processing, (B) Acquiring linear measurements (C) Acquiring chain codes (D) Transforming chain codes to normalized elliptic Fourier descriptors (E) Checking quality and orientation of the outlines (F) Option to open and manage outline descriptors in a text editor. The names of tools and programs used at each step are on the left side of each colored panel. Republished from [30] under a CC BY license, with permission from protocols.io, original copyright 2021.

geometric properties of the spore outlines but enabled a correct outline extraction. Raw and pre-processed images and all information extracted from them are available as a published collection [34].

**Processing shape information.** For smooth spores of *Subulicystidium*, we used elliptic Fourier analysis [35]. It uses an ellipse as a starting shape and searches for the coefficient values that transform the ellipse into the form that reproduces the original outline of the object. These coefficients are called elliptic Fourier descriptors. After normalization, i.e., removing the effects of size, rotation, shift, and starting point of outline recording, they are called normalized elliptic Fourier descriptors (NEFDs). They serve as the continuous numeric variables describing a shape in statistical analysis [1].

We performed elliptic Fourier analysis in the software package SHAPE version 1.3 [36] (http://lbm.ab.a.u-tokyo.ac.jp/~iwata/shape/). Consequently, we used each of the following programs from this package for different analysis steps: ChainCoder, Ch2Nef, NefViewer, PrinComp, and PrinPrint [30]. We started with grayscaling the BMP images in the program ChainCoder (Fig 2C). Then the images were converted to binary (assigned white pixels to spores and black to background) based on a threshold that was mainly automatically selected by ChainCoder and was only rarely adjusted manually. To remove the shape artifacts, an erosion-dilation filter was applied that excludes the noisy pixels from the outline, and rarely also a dilation-erosion filter that fills in the artifact cavities. Then a chain code, i.e., a sequence of x and y coordinates describing the outline, was recorded for each spore.

The chain codes were imported to Ch2Nef program [36] and were checked for being in the same orientation, i.e., with spore hilar appendix placed in the upper left quarter of the image (Fig 2D). To meet this criterion, the chain codes of some spores were rotated to 180 degrees. Ch2Nef then transformed the chain codes into the normalized elliptic Fourier descriptors (NEFDs). As a depth of precision of the contour information, we opted for 20 levels that are called harmonics in the elliptic Fourier analysis [35]. In general, other studies of organism shapes used from 15 to 30 harmonics [5, 37], and using 20 harmonics was sufficient in the classical study from authors of the SHAPE package [36, 38]. As each harmonic is described by four coefficients (= elliptic Fourier descriptors), 80 coefficients characterized each spore's shape. As the first three coefficients are always constant, 77 coefficients served as shape variables for the following analysis step. The approach based on the first harmonic was used to normalize the elliptic Fourier descriptors [36]. We additionally checked that NEFDs are alignable with the NefViewer program (Fig 2E).

We combined NEFDs manually for individual specimens into a single.txt file (Fig 2F). To reduce the multidimensionality of NEFDs data, they were subjected to the principal component analysis (PCA) in the PrinComp program [36]. We used the flexibility of PrinComp and performed in total three principal component analyses (PCAs): i) considering only the symmetric shape variation, ii) considering only the asymmetric shape variation, and iii) considering overall shape variation not differentiated into symmetric and asymmetric (henceforth called global shape variation). As a result of each PCA, those principal components (PC) were retained that explained a substantial amount of variation and were marked by PrinComp as having an eigenvalue >1. Such PCs were called "effective" in PrinComp. Next, the shape variation accounted for by each effective PC was visually inspected with the PrinPrint program [36]. The mean shape plus and minus two times the standard deviation shape for each principal component were reconstructed.

**Processing size information.** For exactly those spores for which NEFDs were obtained, the linear measurements were also taken: length (the longest dimension) and width (the broadest dimension) according to the standard accepted in mycology [8]. Fiji distribution of ImageJ version 1.53c [39] was used as shown in Fig 2B and described in [30]. Fiji does not differentiate whether the measurement is of a category "length" or "width". Therefore, first, the length and then the width of each spore were constantly measured. Though these values were initially placed in a single spreadsheet column, they were split into separate columns of length and width in R

with our protocol [30]. Based on length and width values, we also derived their ratio and used this popular proxy of spore shape as an additional variable in statistical analyses.

## Comparative analyses

While obtaining the shape descriptors and linear measurements, we had control of the number of spores used per specimen and image. However, if the image contained several spores, we could not align the shape descriptors with linear measurements for each particular spore. Even though the input was the same.bmp image, the software for obtaining the shape data (SHAPE) and size data (Fiji) named individual measurements slightly differently. We could overcome it only by producing average-per-image trait values and using them as observations in correlations and discriminant analysis [30]. To calculate the average-per-image trait values, we used regular expressions in R as well as packages "stringr" version 1.4.0 [40] and "dplyr" version 1.3.0 [41].

For comparison, we assigned the morphometric traits to one of the five categories:

i. principal components of symmetric shape variation

ii. principal components of asymmetric shape variation

iii. principal components of global shape variation

iv. linear measurements (length and width)

v. length to width ratio

We explored how similar were all morphometric traits between themselves by Spearman correlations [42]. To answer which morphometric traits allow more efficient automated identification of fungal species, we applied discriminant analysis. We assessed how accurately the individual traits and their combinations could identify (classify) to species level the observations whose species identity is unknown to the statistical model. We found that trait values within species are not necessarily normally distributed and between the species do not have equal variance. For details, see S1 Appendix with the results of Shapiro-Wilk tests (made with R package "RVAideMemoire" version 0.9–79, [43]) and Levene tests (made with R package "heplots" version 1.3–8 [44]). Therefore, we applied an appropriate for such cases flexible discriminant analysis [45] implemented in "mda" R package version 05–2 [46]. The data were randomly split into the train (70%) and test (30%) portions while attempting to balance the presence of records for different species with R package "caret" version 6.0–86 [47]. For cross-validation, splitting the data into train and test portions was repeated 1000 times. The percent of correct identifications (i.e., species assignments) for the test data in each iteration was calculated. Finally, the average identification success rate across iterations was derived using the R code adjusted from [48].

Data were managed in R with the packages "here" version 1.0.1 [49], "conflicted" version 1.0.4 [50], "readr" version 1.4.0 [51], "data.table" version 1.13.4 [52], "dplyr" version 1.3.0 [41], and "report" version 0.2.0 [53]. The results were visualized with the packages „ggplot2"version 3.3.3 [54], "ggpubr" version 0.4.0 [55], function "cor.mtest" [56] and package "corrplot" version 0.84 [57]. Figs 1 and 4 were edited with Inkscape version 0.92.5 [58] and Fig 2 also with Miro online whiteboard (https://miro.com/). The R project with the code, input data and results of analyses is provided in S1 Appendix and as a repository on GitHub (https://github.com/ordynets/size_vs_shape).

## Results

Maximum likelihood phylogenetic analysis showed all ten *Subulicystidium* species on the phylogenetic tree as clearly separate clades, mostly with high bootstrap supports (Fig 3).

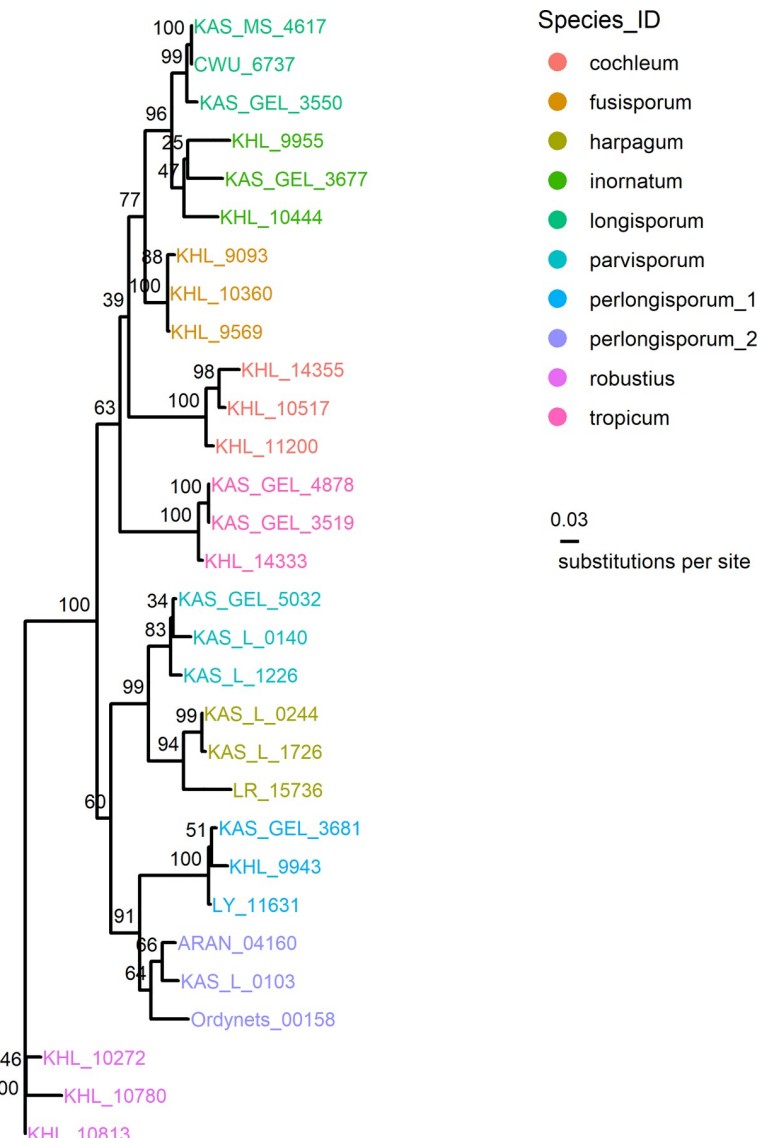

**Fig 3. Phylogenetic analysis resolves 10 *Subulicystidium* species as monophyletic and thus suitable for discriminant analysis.**

After all quality filtering steps, between 10 and 37 spores per specimen were available, totaling 511 spores from 30 specimens. These spores were captured in 401 images, and each image contained one to four spores (S1 Appendix). Therefore we considered 401 average-per-image values as observations for the analyses.

PCA of symmetric shape variation identified only the first axis as effective (capturing 98,7% of the variation in data) while PCA of asymmetric variation three first axes (capturing 83.03, 6.63 and 4.11% of variation), and PCA of global variation two axes (capturing 94.57 and 2.91% of variation). Together with the spore length, spore width, and length to width ratio, this summed up to nine traits which we compared in terms of efficiency for automated species identification (S1 Appendix).

By visually inspecting the shape variation accounted for by each principal component, we found that the 1$^{st}$ PC of symmetric variation reflected well the relative thickness of spores (Fig

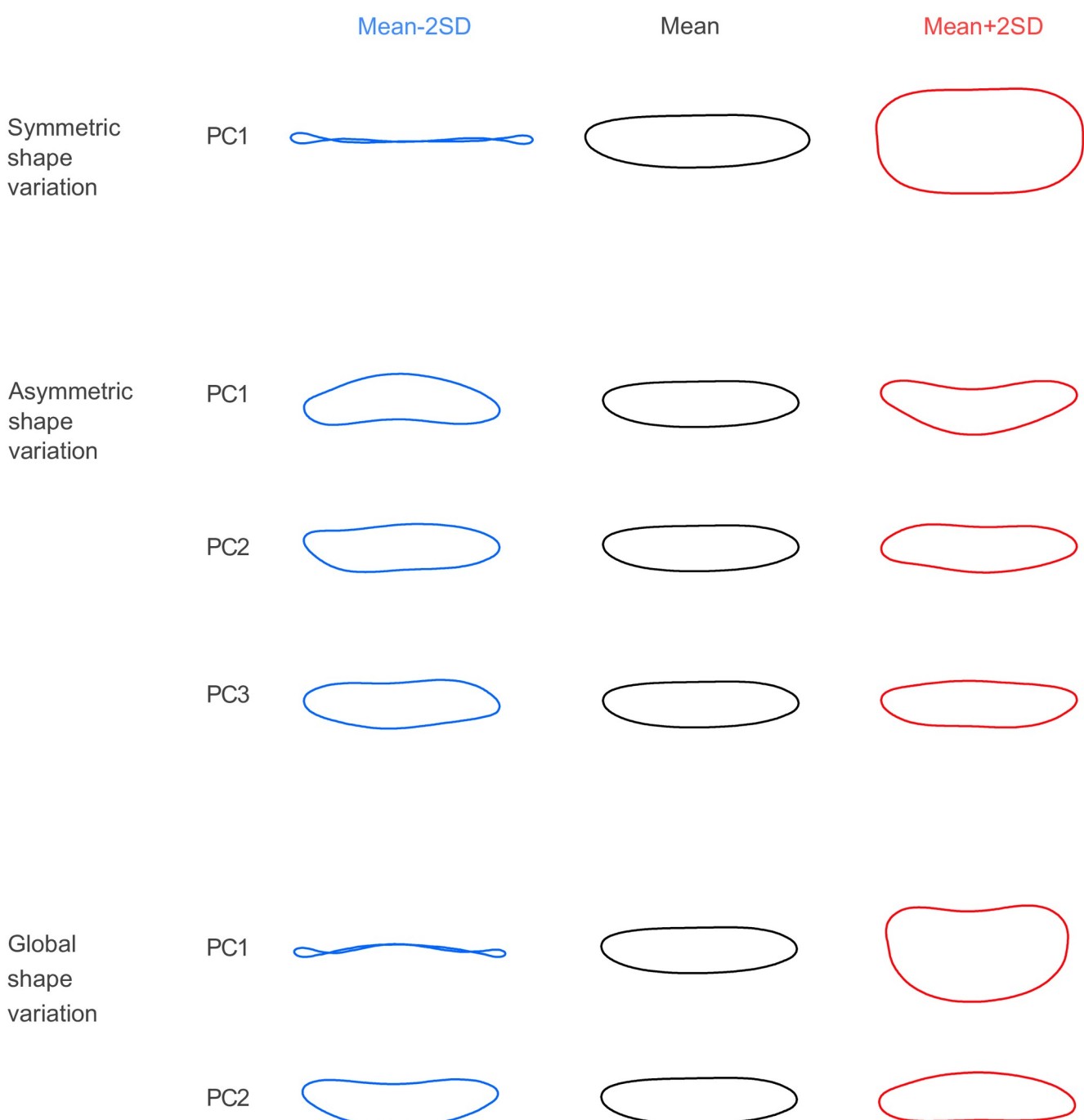

**Fig 4. Symmetric shape variation reflects spore thickness while asymmetric–different aspects of curvature.** The figure shows ranges of shape variation (mean plus and minus two times the standard deviation, SD) that separate principal components (PC) account for.

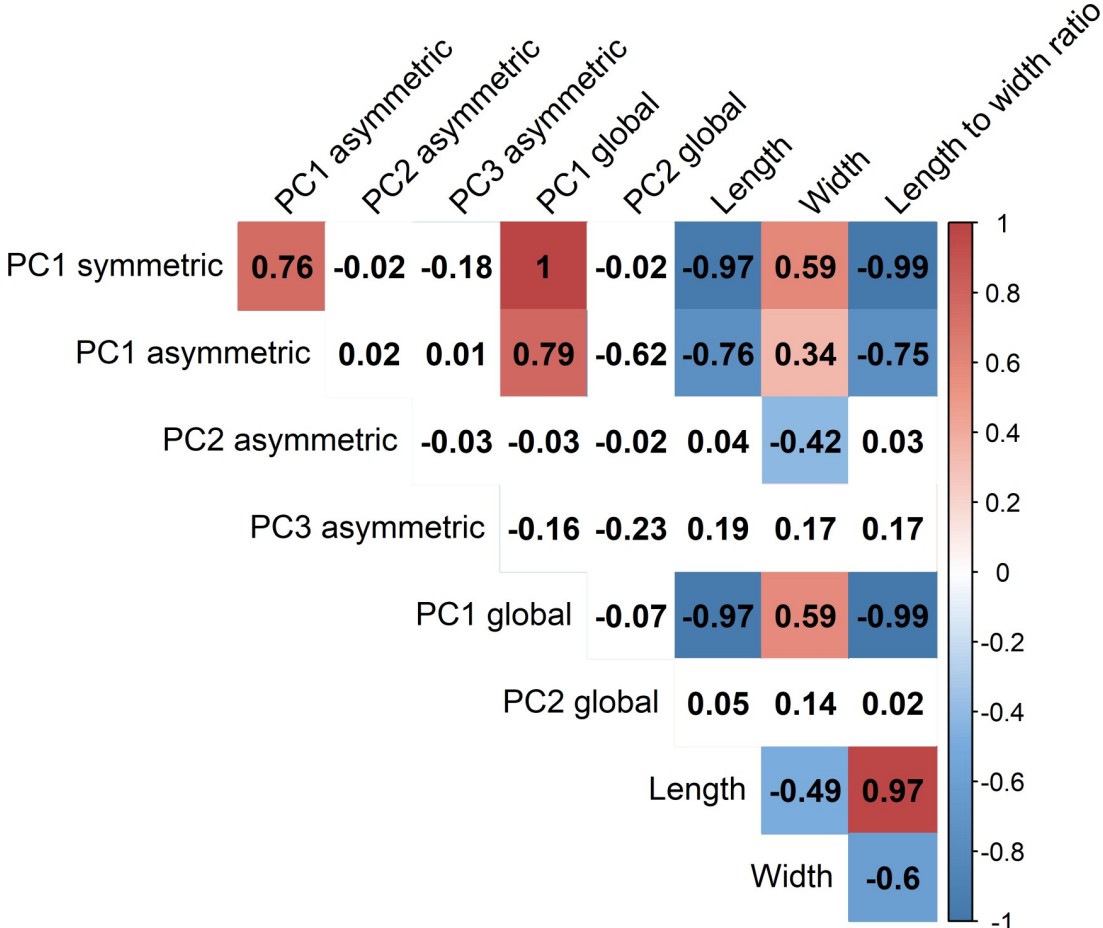

**Fig 5. Main shape descriptors correlate positively between themselves and width and negatively–with the spore length.**
Colors show the direction of Spearman correlations between principal components of shape variation and linear measurements of spores. Only significant correlation values (at level p = 0.05) are colored.

4). The 1st PC of asymmetric shape variation reflected the direction of single curving of the long axis of the spore. The 2nd and 3rd PCs of asymmetric shape variation reflected the curvature of the proximal and distal end of the spore, respectively. The effects of the 1st and 2nd PCs of the global shape variation corresponded to the 1st PC of symmetric and 1st PC of asymmetric shape variation, respectively.

When correlating the individual trait variables, we found that 1st PCs of symmetric, asymmetric, and global shape variation correlated strongly positively with each other (Fig 5). These three variables correlated strongly negatively with the length and moderately positively with the width of the spores. The 2nd PC of asymmetric shape variation correlated moderately negatively with the spore width. In contrast, the 3rd PC of asymmetric shape variation and the 2nd PC of the global shape variation did not correlate significantly with any other trait. Length and width correlated moderately negatively between themselves, and each correlated very differently with the length to width ratio. However, the correlation was stronger between the length and length to width ratio.

In the discriminant analysis, the species identification success rate for the individual trait group was highest for the global shape variation (61.5%, model G, Fig 6). In comparison, it was slightly lower for the length combined with width (59.1%, model LW). Symmetric shape

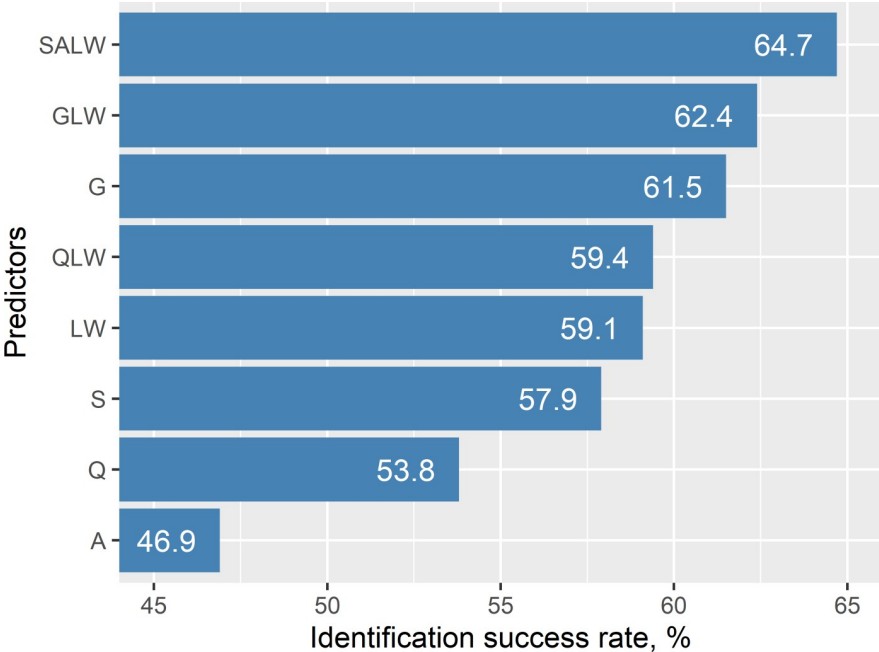

**Fig 6. Discriminant models including both size and shape descriptors outperform those with either predictor type.** For each value on y axis, species identification success rate from discriminant analysis is displayed as an average value from 1000 replicates of cross-validation. Abbreviations on y axis: S = symmetric shape variation (first principal component), A = asymmetric shape variation (three first principal components), G = global shape variation (two first principal components), LW = length + width, Q = length to width ratio, QLW = length to width ratio + length + width, GLW = global shape variation + length + width, SALW = symmetric shape variation + asymmetric shape variation + length + width.

variation identified the fungal species better than the length to width ratio (57.9%, model S, vs. 53.8%, model Q). The asymmetric shape variation resulted in the lowest identification success rate, viz. 46.9% (model A). The model that combined symmetric, asymmetric shape variation, and linear measurements achieved the highest identification success rate (64.7%, model SALW). Combining the global shape variation with linear measurements provided a slightly lower identification success rate (62.4%, model GLW).

## Discussion

Geometric morphometric analysis of shapes helps to describe details of organisms and understand their interaction with the environment. However, it has been unclear how much geometric information on the outline can improve taxa identification when combined with traditional linear measurements. Using fungal spores as an example, we confirmed the hypothesis that combining geometric information on the outline with linear measurements provides better results of automated species identification than either source of evidence alone does. We also showed that, for species identification, geometric information outperformed the classical length to width ratio. Our results align with the study where another approach, landmark-based geometric morphometrics, was combined with traditional morphometrics [11].

Concerning fungal spores, whose morphology is highly relevant for fungal taxonomy, geometric morphometrics was rarely applied so far. Besides the fact that geometric morphometrics is a young discipline with roots in the vertebrate zoology, this is probably explained by the small size of the fungal spores (mostly under 50 μm and often under 10 μm [59]) and related difficulties of imaging. Nevertheless, we showed that it is possible to adequately extract

and analyze the geometric shape information from microscopy images of fungal spores on a routine basis.

For extracting shape information, we opted for the program package SHAPE [36]. It allows extracting several outlines from the same image, has flexibility for performing PCA, and convenient interface. On the other hand, we had to switch between the separate programs for different steps, meaning handling several outcomes and an extra effort in data management. Furthermore, the graphical user interface of SHAPE (or any other GUI program) means investing massive time effort in repeating the analysis and increased risk of producing an error. Therefore, it is advisory to work in a single environment with all analysis steps written as a code in the future. Running analyses from a code will simplify the data management and enhance the reproducibility of the research. Functions in R language designed by [4] and further developed by [60] and [61] are promising for this.

With our study design, we observed high correlations between several trait variables. The most striking is the correlation between the shape descriptors (PC1 of symmetric and asymmetric variation) and size descriptors (especially length). It is likely due to our choice to represent the size as traditional linear measurements known to co-vary with shape [2]. We also generated the size variable that integrates separate linear measurements and is less dependent on shape (square root of the length time width, as one possible option according to [4]). This size variable indeed correlated weaker with shape variables. Interestingly, the discriminant model based on this size descriptor predicted the species worse than the model based on the length added to width (38.9% vs. 59.1%, see S1 Appendix). This result further demonstrates that traditional linear measurements incorporate information on the size as well as shape [2]. In the future, taxonomists may gain additional information when implying size descriptors that are mathematically completely independent (orthogonal) from shape [4].

While linear measurements predicted species correctly in 59.1% cases (LW model), adding shape descriptors increased the prediction rate to 64.7% (SALW model). The increase of nearly 5% may first appear as not prominent. We explain this pattern by including in the analyses the species with very different spores (e.g., from needle-shaped to short-cylindric). It made the weight of the first PCs (of symmetric and asymmetric variation) very high and caused them to correlate strongly with the linear measurements. A strong correlation of the first PC of shape variation with the linear size is a known phenomenon in geometric morphometrics [2]. In our case, it could have contributed to overfitting the discriminant model and result in only a slight increase of success of identification in the SALW model compared to LW model. If the range of spore shape variation between the taxa were narrower, we expect the first PC to correlate weaker with the linear measurements and result in higher identification success rates in the SALW model. In the simulation with removing the 1st PC of global variation from PCA scores of NEFD data, we found the subsequent PCs a bit more equalized in the amount of reflected variation (see S1 Appendix). In general, the success of automated identification depends on the range of variation in the data [62]. Therefore, we argue it is essential not to focus on a bold number but rather to be able to disentangle the contribution of separate descriptors to the discriminant model. We are looking forward to seeing the results of discriminant analyses when applying our approach to other microscopic objects with different ranges of morphological variation.

The identification success rate in our discriminant analyses reached the maximum value of 64.7%. It is possible to compare only roughly our results with other studies on microscopic structures, where different traits and/or different classification methods were used. An identification success rate of 66.2% was achieved for seven traits and a higher rate (74.4%) for 20 traits of fungal spores (mostly size descriptors) [63]. Similar to our results were obtained in discriminant analyses of size and texture traits in diatom algae (55.1%) [64]. Nearly 100% correct

identifications of diatom algae were possible based on shape and texture descriptors, although an additional classification method was applied above the discriminant analysis [65].

In another morphometric analysis, i.e., of statolith shapes in Cubomedusae, the identification success rate was highly dependent on the number of observations per group [33]. Due to technical reasons explained in the Methods section, we performed analyses on the average-per-image data. We admit that working with spore-level data would provide a more precise estimate of the correct identification rate. To include ten species and three specimens for each species into analyses, we acquired images of spores from squash preparations of dried herbarium specimens collected between 1978 and 2016. Such spore samples may include unmature spores that can affect measurements results [66]. We believe we could essentially minimize this risk due to our previous fungal taxonomy experience and merely by considering floating (not attached to basidia) basidiospores. Squash preparations from herbarium specimens do not always provide the same number of well-preserved spores. We were able to measure 10 and 37 spores per specimen while it was recommended to measure at least 10 spores but better 30 for the study of spore size [8]. We counteracted the sample size difference by the design of discriminant analysis. There, we took a subset of 70% of measurements for each species for building the training data. Furthermore, in each of 1000 iterations of cross-validation, different subsamples of spores per species were drawn for building the training and test data. For future studies, we recommend working with exclusively mature spores from spore prints as implemented in [66] and ensure an equal sample size as in [8, 67]. Additionally, possible measurements biases from measuring fresh vs. dried spores and spores in different mounting media should be kept in mind [10, 66, 68].

Future work on quantitative shape analysis of microscopic objects should cope with several challenges. These objects may have very different outline properties in various projections. In the current study, we focused on the spores' lateral face, which is more peculiar in *Subulicystidium* and allows to capture the curvature along the spores' long axis (while the frontal face masks this feature). However, in spores of other fungal taxa, the lateral and frontal projections are both important [29], and their shapes should be analyzed with the same attention. Combining three projections of statoliths maximized the identification success rate in Cubomedusae [33]. On the other hand, there are numerous cases (e.g., fungi from the phylum Ascomycota) with radially symmetric spores formed inside sporangia and lacking any attachment loci. In this case, selecting the spores for outline extraction and processing digitized outlines will be more straightforward.

While our study focused on smooth microscopic objects, they may be supplemented by additional projections like warts or ridges or distinct spines in other taxa. It remains unexplored how correctly ornamentations could be captured by the elliptic Fourier method, which tends to downweight such elements [1]. The ornamented spores of fungi may become an interesting object for accommodating landmark-based and semilandmark-based methods of geometric morphometrics for microscopic objects [69]. Furthermore, these methods are extendable to the three-dimensional analysis. The latter is becoming feasible even for objects smaller than 10 μm due to nano-computed tomography [70].

Few attempts were made to link dispersal and deposition properties and a nutritional mode of fungi with the shape of their spores. However, the shape was expressed as a categorical variable [71] or as a length to width ratio that provides limited geometric information [72]. Therefore, we believe that including quantitative shape descriptors from the geometric morphometric analysis will provide a deeper understanding of the relationship between morphological and ecological traits in fungi.

We conclude that adding geometric information on the outline to linear measurements improves the identification of the organisms. It also opens new possibilities for testing

evolutionary and ecological hypotheses that link morphology and functions. We provide an open access protocol to propagate a broader use of geometric morphometric analysis for microscopic propagules of fungi and other organisms and stimulate the development of more efficient solutions for addressing challenges of radial asymmetry and ornamented surface.

## Supporting information

**S1 Table. GenBank accessions of used DNA sequences and metadata on voucher specimens.**
(TSV)

**S1 Appendix. R project with the code, input data, and results of analyses.**
(ZIP)

## Acknowledgments

We acknowledge the curators of fungal collections in ARAN, CWU, GB, LY, and O for providing specimens for our study, and personally, Karl-Henrik Larsson, Janett Riebesehl, and Manuel Striegel, who collected most of the specimens. David Scherf, Ludmila Lysenko, Ilka Kellner, Robert Liebisch, and Jonathan Denecke performed the DNA lab work and helped with the microscopy of the collections. Anton Savchenko and Oleh Prylutskyi provided feedback on the protocol for morphometrics, and Iryna Yatsiuk has kindly copyedited the earlier version of the manuscript. Ladislav Hodac is thanked for introducing the first author to outline-based geometric morphometrics and program package SHAPE.

## Author Contributions

**Conceptualization:** Alexander Ordynets.

**Data curation:** Alexander Ordynets, Sarah Keßler.

**Formal analysis:** Alexander Ordynets.

**Investigation:** Alexander Ordynets, Sarah Keßler.

**Methodology:** Alexander Ordynets, Sarah Keßler.

**Project administration:** Alexander Ordynets.

**Resources:** Ewald Langer.

**Software:** Alexander Ordynets.

**Supervision:** Alexander Ordynets.

**Validation:** Alexander Ordynets, Sarah Keßler.

**Visualization:** Alexander Ordynets, Sarah Keßler.

**Writing – original draft:** Alexander Ordynets.

**Writing – review & editing:** Alexander Ordynets, Ewald Langer.

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
