## [Decision Letter · Decision Letter 0]

18 May 2021

PONE-D-21-10269

Quantitative analysis of spore shapes improves identification of fungi

PLOS ONE

Dear Dr. Ordynets,

Thank you for submitting your manuscript to PLOS ONE. After careful consideration, we feel that it has merit but does not fully meet PLOS ONE’s publication criteria as it currently stands. Therefore, we invite you to submit a revised version of the manuscript that addresses the points raised during the review process.

The whole manuscript deserves greater authors' attention towards clarifying major aims of the study and increasing the overall academic impact. Figures should also be revised as per the Reviewers' reports. Moreover, authors should put additional effort in consulting the corresponding scientific literature in order to sustain their claims as well as to compare their results with other studies. The mansucript would also benefit from copyediting by a native English speaker in order to polish the language and to increase readability.

We look forward to receiving your revised manuscript.

Kind regards,

Branislav T. Šiler, Ph.D.

Academic Editor

PLOS ONE

Journal Requirements:

2. We noted in your submission details that a portion of your manuscript may have been presented or published elsewhere.

"Figure 2, exactly as present in this manuscript, was also included into our recently published open access protocol "Ordynets A, Keßler S, Willemsens C (2021) Extracting shape and size information from fungal spores. protocols.io " ext-link-type="uri" xlink:type="simple">https://dx.doi.org/10.17504/protocols.io.bdeii3ce", where it was published under the license Attribution 4.0 International (CC BY 4.0). The figure allows to undestand better both online protocol and current research paper. It's licensing type (CC BY) allows us to use it in this mansuscript.

The source images, on which all measurements are based, were published as a collection and registered with a DataCite DOI:  "Ordynets A, Lysenko L, Kellner I, Scherf D, Liebisch R, Denecke J, et al. Data for the study “Quantitative analysis of spore shapes improves identification of fungi.” Universität Gesamthochschule Kassel; 2021. doi:10.15156/BIO/807451". The size of this data (1.4 GB) did not allow to include them as a supporting information into this manuscript."

Reviewers' comments:

Reviewer's Responses to Questions

**Comments to the Author**

1. Is the manuscript technically sound, and do the data support the conclusions?

Reviewer #1: Yes

Reviewer #2: Yes

2. Has the statistical analysis been performed appropriately and rigorously? 

Reviewer #1: Yes

Reviewer #2: Yes

3. Have the authors made all data underlying the findings in their manuscript fully available?

Reviewer #1: Yes

Reviewer #2: Yes

4. Is the manuscript presented in an intelligible fashion and written in standard English?

Reviewer #1: Yes

Reviewer #2: No

5. Review Comments to the Author

Reviewer #1: The manuscript "Quantitative analysis of spore shapes improves identification of fungi" is technically sound and all data support the conclusions. Applied methodology is very innovative in the field of mycology, especially because it is used for the purposes of distinguishing different fungal species more easily and correctly. In my opinion statistical analysis is performed appropriately. Authors ensured that all data from the manuscript are available to the potential readers. English is quite good but I would suggest additional final proofing by a native English speaker if possible. As well, I would suggest that you consider my comments and apply necessary changes to improve the manuscript. My highest concern is that you didn't find many mycological articles dealing with spore shapes even though there are many of them. I proposed some of them for you to read and cite if you find them appropriate (I think they are).

Reviewer #2: The authors should be commended for using NEFD to study spore morphology - it is an up and coming method for which there are few studies focused on fungi. The authors' procedural and statistical approaches are thorough and appropriate.

The manuscript would benefit from a greater narrative. The methods are very strong but they get lost without greater intellectual context. It was often unclear why this study was conducted, why the particular genus was selected, and what is the broader scientific question. I found the hypotheses to be overly presumptuous - one employs NEFD precisely to answer the sorts of questions posed by the three hypothesis the authors give; so asking if NEFD works for that purpose, when it obviously does (otherwise the authors wouldn't have chosen to use NEFD) seems circular. I was especially confused by Figure 6 - I had trouble understanding what that figure represents and how it was calculated. If the authors mean to show that NEFD can be used to predict species identity from spore shape I would be very surprised by that claim given that almost all of the shape variation could be explained by the first PC for symmetrical and asymmetrical analyses (which seemed to correspond to features that are not unique to Subulicystidium and I am doubtful that this method would work beyond the idiosyncrasies of the dataset the authors employed).

I thought the figures need a lot of work and greater explanation in the legends. I would have liked to have seen more explanation on the imaging protocols besides a citation to past work, as well as an explanation as to why different numbers of images were used per species and how sample size differences were handled.

Linguistic usage use also requires a fair amount of work, especially in the introduction and conclusion.

6. PLOS authors have the option to publish the peer review history of their article (what does this mean?). If published, this will include your full peer review and any attached files.

Reviewer #1: No

Reviewer #2: No

---

## [Author Response · Author response to Decision Letter 0]

14 Jul 2021

We thank Editor and Reviewers for numerous comments which allowed us to substantially improve the manuscript. We address all comments in details in file "Response to Reviewers".

---

## [Editor Report · Decision Letter 1]

26 Jul 2021

Geometric morphometric analysis of spore shapes improves identification of fungi

PONE-D-21-10269R1

Dear Dr. Ordynets,

We’re pleased to inform you that your manuscript has been judged scientifically suitable for publication and will be formally accepted for publication once it meets all outstanding technical requirements.

Kind regards,

Branislav T. Šiler, Ph.D.

Academic Editor

PLOS ONE
---

## [Editor Report · Acceptance letter]

28 Jul 2021

PONE-D-21-10269R1 

Geometric morphometric analysis of spore shapes improves identification of fungi 

Dear Dr. Ordynets:

I'm pleased to inform you that your manuscript has been deemed suitable for publication in PLOS ONE. Congratulations! Your manuscript is now with our production department. 

Kind regards, 

on behalf of

Dr. Branislav T. Šiler 

Academic Editor

PLOS ONE